# VITA-1.5: Towards GPT-4o Level Real-Time Vision and Speech Interaction

Chaoyou Fu[1,2,♠], Haojia Lin[4], Xiong Wang[3], Yi-Fan Zhang[5], Yunhang Shen[3]
Xiaoyu Liu[1,2], Haoyu Cao[3], Zuwei Long[3], Heting Gao[3], Ke Li[3], Long Ma[3],
Xiawu Zheng[4], Rongrong Ji[4], Xing Sun[3,†], Caifeng Shan[1,2], Ran He[5]

[1] State Key Laboratory for Novel Software Technology, Nanjing University
[2] School of Intelligence Science and Technology, Nanjing University
[3] Tencent Youtu Lab, [4] XMU, [5] CASIA
♠ Project Leader   † Corresponding Author

Demo Video: Click YouTube Link

Source Code: `https://github.com/VITA-MLLM/VITA`

## Abstract

Recent Multimodal Large Language Models (MLLMs) have typically focused on integrating visual and textual modalities, with less emphasis placed on the role of speech in enhancing interaction. However, speech plays a crucial role in multimodal dialogue systems, and implementing high-performance in both vision and speech tasks remains a challenge due to the fundamental modality differences. In this paper, we propose a carefully designed multi-stage training methodology that progressively trains LLM to understand both visual and speech information, ultimately enabling fluent vision and speech interaction. Our approach not only preserves strong vision-language capacity, but also enables efficient speech-to-speech dialogue capabilities without separate ASR and TTS modules, significantly accelerating multimodal end-to-end response speed. By comparing against state-of-the-art counterparts across benchmarks for image, video, and speech, we demonstrate that our **omni** model is equipped with both strong visual and speech capabilities, making omni understanding and interaction.

## 1 Introduction

Recent advancements in MLLMs [1, 2, 3, 4, 5, 6, 7, 8, 9] have led to significant progress, particularly in integration of visual and textual modalities. The introduction of visual information into LLMs has notably enhanced model capabilities across various multimodal tasks. However, with the growing appeal of human-computer interaction, the role of the speech modality has become increasingly prominent, especially in multimodal dialogue systems. In such a system, speech not only serves as a key medium for information transmission but also greatly improves the naturalness and convenience of interactions. Consequently, integrating visual and speech modalities to achieve multimodal interactions has emerged as a critical research focus.

The integration of vision and speech in MLLMs is not straightforward due to their inherently differences [10]. For example, visual data, such as images, convey spatial information, while speech data convey dynamic changes in time series. These fundamental differences pose challenges for simultaneous optimization of both modalities, often leading to conflicts during training. For

39th Conference on Neural Information Processing Systems (NeurIPS 2025).

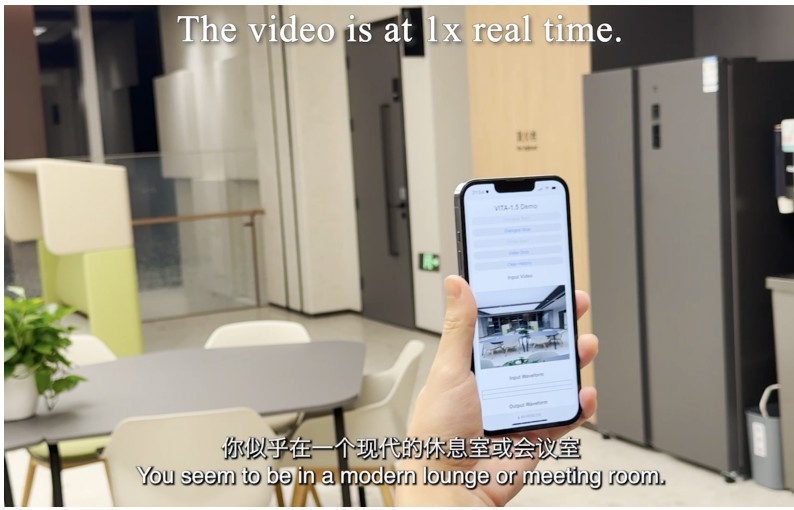

Figure 1: VITA-1.5 enables near real-time vision and speech interaction via an end-to-end framework. It allows you to turn on the camera and have a fluent speech conversation. **Please see our demo video at this YouTube link.**

instance, the inclusion of speech data may degrade performance on vision tasks, and vice versa. In addition, traditional speech-to-speech systems rely on separate modules for Automatic Speech Recognition (ASR) and Text-to-Speech, which can increase latency and reduce coherence, limiting their practicality in real-time applications [11, 12, 13, 14, 15].

In this paper, we introduce VITA-1.5, a multimodal LLM that integrates vision, language, and speech through a carefully designed three-stage training methodology. The training strategy progressively incorporates vision and speech data, relieving modality conflicts while maintaining strong multimodal performance. In the first stage, we focus on vision-language by training visual adapters and fine-tuning the model with descriptive caption and visual QA data. This step establishes the model's foundational visual capabilities, enabling robust image and video understanding. The second stage introduces audio input processing by training an audio encoder using speech-transcription paired data, followed by fine-tuning with speech QA data. This stage equips the model with the ability to understand and respond to audio inputs effectively. Finally, in the third stage, we train an audio decoder to enable end-to-end speech output, eliminating the need for external TTS modules. This allows VITA-1.5 to generate fluent speech replies, enhancing the naturalness and interactivity of multimodal dialogue systems.

We have conducted extensive evaluations on various benchmarks related to image, video, and speech understanding, comparing the results with both open-source and proprietary models. VITA-1.5 demonstrates comparable perception and reasoning capabilities comparable to leading image/video based MLLMs, and shows significant improvements in the speech capability.

## 2 Related Work

Recently, thanks to the rapid development of language models such as GPTs [16, 17], LLaMA [18, 19], Alpaca [20], Vicuna [21], and Mistral [22], researchers have successfully extended text comprehension to multimodal understanding/reasoning through techniques like multimodal alignment and instruction tuning. For example, models such as LLaVA [1], Qwen-VL [23], Cambrian-1 [24], Mini-Gemini [25], MiniCPM-V 2.5 [26], DeepSeek-VL [27], and SliME [28] have made significant advances in image perception and reasoning, while models like LongVA [29] and Video-LLaVA [30] have showcased the latest progress in video understanding. These models are increasingly capable of handling diverse data types, driving the continuous improvement of multimodal perception and understanding capabilities.

Beyond visual modalities, recent years have also witnessed significant progress in incorporating speech capabilities into LLMs, driven by the increasing demand for natural human-computer interaction. The dominant approach has been to cascade ASR, LLM, and TTS modules. This text-centric

approach faces fundamental limitations due to the loss of paralinguistic features like tones and emotions. While works like [31] and [32] have attempted to address these issues by incorporating speech encoders and emotion vectors, they still rely on speech transcription, resulting in substantial latency issues that impact the user experience. The emergence of proprietary models like GPT-4o [33] has demonstrated the possibility of end-to-end speech interaction, inspiring a new wave of research in speech-enabled MLLMs. Following this trend, several notable works have emerged in the open-source community. Models such as Mini-Omni2 [34], LLaMA-Omni [35], and Moshi [36] have explored various strategies for aligning speech modality with LLMs and achieving duplex dialogue capabilities. While these open-source efforts have successfully enabled duplex speech interaction with LLMs, they still lack the capability to handle visual modalities as demonstrated by GPT-4o, limiting their applications in scenarios requiring both visual and speech understanding.

Despite these advances in both visual and speech modalities, a significant gap remains between proprietary and open-source models. Compared to proprietary models that support multiple modalities, including audio, image, and text, e.g., GPT-4o [37] and Gemini-Pro 1.5 [38], most open-source models have primarily focused on image and text modalities [2]. Moreover, few open-source models have involved multimodal interaction capabilities, which is a relatively unexplored area. While works like VITA-1.0 [12] have made initial attempts to introduce speech for human-computer interaction, introducing additional speech data poses challenges to the model's original multimodal abilities. Furthermore, speech generation typically relies on existing TTS systems, which often results in high latency, thus impacting user experience. In this paper, we present VITA-1.5 that leverages refined training strategies, excelling in perceiving data across four modalities (video, image, text, and audio), while also realizing near real-time vision and speech interaction.

## 3 VITA-1.5

### 3.1 Model Architecture

The overall architecture of VITA-1.5 is depicted in Fig. 2. The input side is the same as that of the VITA-1.0 version [12], that is, adopting the configuration of "Multimodal Encoder-Adaptor-LLM". It combines the Vision/Audio Transformer and the Multi-Layer Connector with an LLM for joint training, aiming to enhance the unified understanding of vision, language, and audio. With respect to the output side, VITA-1.5 has its own end-to-end speech module, instead of using the external TTS model like the original VITA-1.0 version.

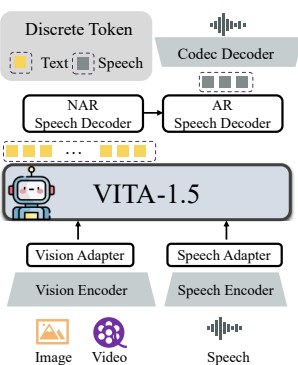

Figure 2: **Overall Architecture of VITA-1.5**. The input side consists of vision and audio encoders, along with their adapters. The output side has an end-to-end speech generation module, rather than directly using an TTS model.

#### 3.1.1 Visional Modality

**Visual Encoder.** VITA-1.5 adopts InternViT-300M[1] as the visual encoder, with an input image size of 448×448 pixels, generating 256 visual tokens per image. For high-resolution images, VITA-1.5 employs a dynamic patching [39] strategy to capture local details, improving the accuracy of image understanding.

**Video Processing.** Videos are treated as a special type of multiple-image input. If the video length is shorter than 4 seconds, 4 frames are uniformly sampled; for videos between 4 and 16 seconds, one frame per second is sampled; for videos longer than 16 seconds, 16 frames are uniformly sampled. No dynamic patching is applied to video frames to avoid excessive visual tokens that could hinder processing efficiency.

**Vision Adapter.** A two-layer MLP is used to map the visual features to visual tokens suitable for the subsequent understanding of LLM.

---

[1] https://huggingface.co/OpenGVLab/InternViT-300M-448px

### 3.1.2 Audio Modality

**Speech Encoder.** Similar to [40], our audio encoding module consists of multiple downsampling convolutional layers (4x downsampling) and 24 Transformer blocks (with a hidden size of 1024). The downsampling layers help reduce the frame rate of the audio features, improving the processing speed of LLM. The audio encoder has about 350M parameters and an output frame rate of 12.5Hz. Mel-filter bank features are used as the input of the audio encoder, with a window size of 25ms and a shift of 10ms [40].

**Speech Adapter.** It consists of multiple convolutional layers with 2x downsampling.

**Speech Decoder.** TiCodec [41] is used as our codec model, customizing a single codebook with a size of 1024. This single-codebook design simplifies the decoding process during the inference phase. The codec model is responsible for encoding continuous speech signals into discrete speech tokens with the frequency of 40Hz, and at the same time has the ability to decode them back into speech signals with the sample rate of 24,000Hz.

The current LLM can only output text tokens, and the speech generation capability requires the LLM to be able to output speech tokens. To this end, we add two speech decoders after the text tokens following [40]: 1) **Non-Autoregressive (NAR) Speech Decoder**, which processes text tokens globally and models semantic features, with the aim of generating an initial distribution of speech tokens; 2) **Autoregressive (AR) Speech Decoder** generates higher quality speech tokens step by step, based on the speech information produced by the NAR decoder. The final sequence of speech tokens is then decoded into a continuous speech signal flow (waveform) using the speech decoder of the Codec model. We adopt 4 LLaMA decoder layers for both NAR and AR speech decoders, where the hidden size is 896 and the parameter size is about 120M.

### 3.2 Training Data

As shown in Table 1, the training data of multimodal instruction tuning encompass a wide range of categories, such as caption data and QA data, both Chinese and English. During different training phases, subsets of the overall dataset are selectively sampled to serve different objectives. Specifically, the datasets are categorized as follows:

- **Image Captioning Data.** Datasets such as ShareGPT4V [42], ALLaVA-Caption [43], SharedGPT4o-Image[2], and synthetic data are used to train the model to generate descriptive languages for images.

- **Image QA Data.** Datasets like LLaVA-150K[3], LLaVA-Mixture-sample [1], LVIS-Instruct [44], ScienceQA [45], ChatQA [46], and subsets sampled from LLaVA-OV [47], such as general image QA and mathematical reasoning datasets, are utilized to train the model in answering image-based questions and performing visual reasoning tasks.

- **OCR & Diagram Data.** This category supports the model in understanding OCR and diagram content, using datasets such as Anyword-3M [48], ICDAR2019-LSVT[4], UReader [49], SynDOG[5], ICDAR2019-LSVT-QA[6], and corresponding data sampled from LLaVA-OV.

- **Video Data.** Datasets like ShareGemini [50] and synthetic data are used to train the model to handle video inputs and perform tasks such as captioning and video-based QA.

- **Pure Text Data.** This category enhances the model's capability to understand and generate languages, facilitating text-based QA tasks.

In addition to the image and video data listed in Table 1, 110,000 hours of internal speech-transcription paired ASR data, covering both Chinese and English, are incorporated to train the audio encoder and align the audio encoder with the LLM. Furthermore, 3,000 hours of text-speech paired data generated by a TTS system are used to train the speech decoder.

---

[2] https://sharegpt4o.github.io/
[3] https://huggingface.co/datasets/liuhaotian/LLaVA-Instruct-150K
[4] http://icdar2019.org/
[5] naver-clova-ix/synthdog-en
[6] http://icdar2019.org/

Table 1: Training data of multimodal instruction tuning. The images of the synthetic data come from open-source datasets like Wukong [51], LAION [52], and CC12M [53].

| Data Scenario | QA Type | Dataset Name | Questions (K) | Language |
|---|---|---|---|---|
| General Image | Description | ShareGPT4V | 99.50 | Eng |
| | | ALLaVA-Caption | 697.40 | Eng |
| | | ShareGTP4o-Image | 55.50 | Eng |
| | | Synthetic Data | 593.70 | CN |
| | QA | LLaVA-150K | 218.36 | CN |
| | | LLaVA-Mixture-sample | 1872.10 | Eng |
| | | LVIS-Instruct | 939.36 | Eng |
| | | ScienceQA | 12.72 | Eng |
| | | ChatQA | 7.39 | Eng |
| | | LLaVA-OV General | 1754.65 | Eng |
| | | LLaVA-OV Math Reasoning | 1140.92 | Eng |
| | | Synthetic Data | 212.68 | CN |
| OCR & Diagram | Description | Anyword-3M | 1709.30 | CN |
| | | ICDAR2019-LSVT | 366.30 | CN |
| | | UReader | 100.00 | Eng |
| | | SynDOG-EN | 100.00 | Eng |
| | | SynDOG-CN | 101.90 | CN |
| | QA | ICDAR2019-LSVT-QA | 630.08 | CN |
| | | LLaVA-OV Doc Chart Screen | 4431.50 | Eng |
| | | LLaVA-OV General OCR | 404.20 | Eng |
| General Video | Description | ShareGemini | 205.70 | CN |
| | | Synthetic Data | 569.40 | CN & Eng |
| | QA | Synthetic Data | 4336.30 | CN & Eng |
| Pure Text | QA | Synthetic Data | 1574.20 | CN & Eng |
| | Total | | 22133.16 | CN & Eng |

## 3.3 Three Stage Training Strategies

In order to ensure that VITA-1.5 performs well in tasks involving vision, language, and audio, we have to face a key challenge, i.e., training conflicts between different modalities. For example, adding the speech data could negatively impact the understanding of the vision data, as the features of speech differ significantly from those of vision, causing interference during the learning process. To address this challenge, we devise a three-stage training strategy as shown in Fig. 3. The core idea is to gradually introduce different modalities into the model, allowing it to increase the power of a new modality while maintaining the power of the existing modalities.

### 3.3.1 Stage 1: Vision-Language Training

**Stage 1.1 Vision Alignment.** In this stage, our goal is to bridge the gap between vision and language. The features of the former are extracted from the pre-trained vision encoder InternViT-300M, and the latter is introduced through the LLM. We use 20% of the descriptive caption data from Table 1 for training, where only the visual adapter is trainable, while the other modules are frozen. This approach allows the LLM to initially align the visual modality.

**Stage 1.2 Vision Understanding.** In this stage, our goal is to teach the LLM to transcribe image content. Toward this end, we use all the descriptive caption data from Table 1. During this process, the encoder and adapter of the visual module, as well as the LLM, are trainable. The focus is to enable the model to establish a strong connection between vision and language by learning from descriptive texts about images, allowing it to understand image content via generating natural language descriptions.

**Stage 1.3 Vision SFT.** Following Stage 1.2, the model has acquired a basic understanding of images and videos. However, the instruction following ability is still limited, and it is difficult to cope with the visual QA task. To achieve this, we use all the QA data from Table 1 while retaining 20% of the descriptive caption data to increase the diversity of the dataset and the complexity of the tasks.

During training, the encoder and adapter of the visual module, as well as the LLM, are trainable. The key objective of this stage is to enable the model not only to understand visual content but also to answer questions following instructions.

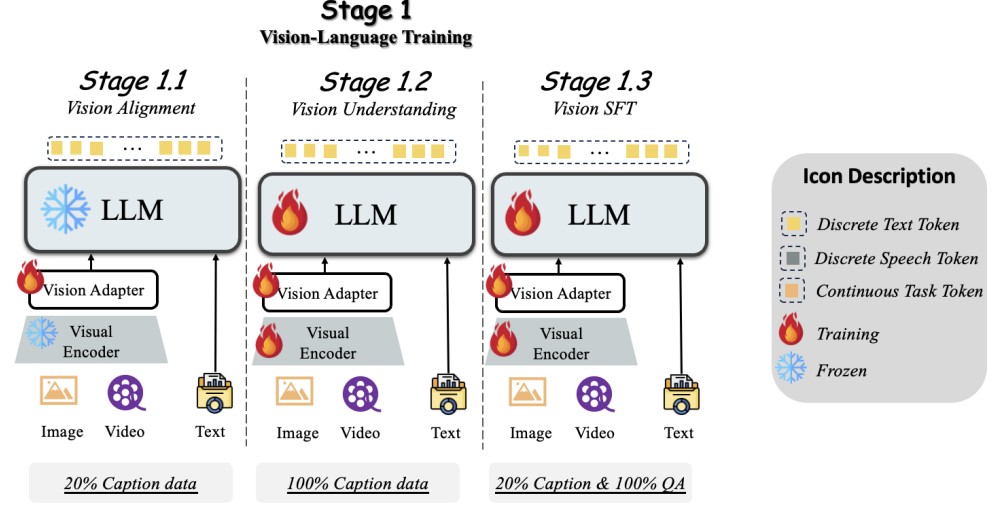

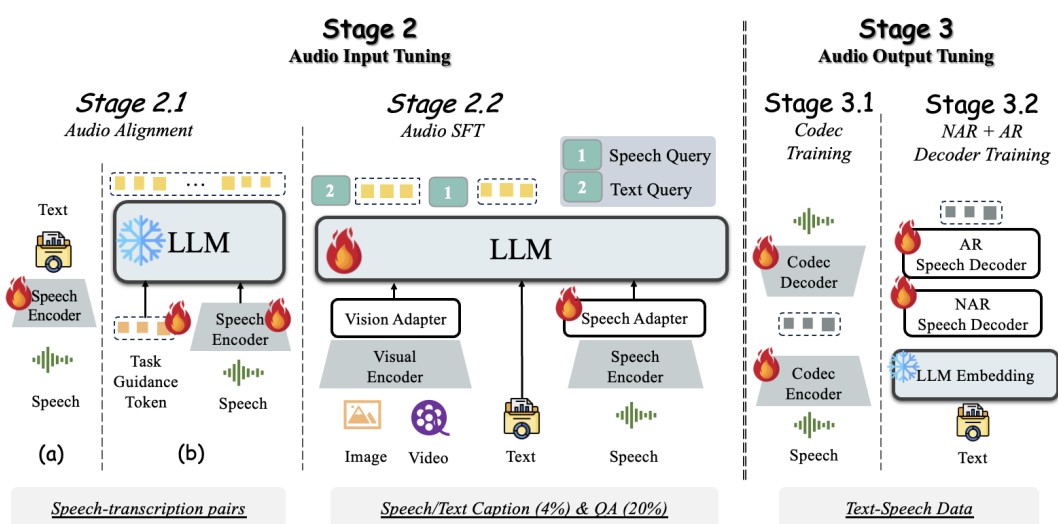

Figure 3: **Training Pipeline of VITA-1.5**. The training process is divided into three stages to incrementally incorporate vision and audio into the LLM while relieving modality conflicts. Stage I focuses on **Vision-Language Training**, including vision alignment (Stage 1.1, using 20% caption data from Table 1), vision understanding (Stage 1.2, using 100% caption data), and instruction tuning for visual QA (Stage 1.3, using 20% caption data and 100% QA data). Stage 2 introduces **Audio Input Tuning**, with audio alignment (Stage 2.1, utilizing 11,000 hours of speech-transcription pairs) and instruction tuning for speech QA (Stage 2.2, sampling 4% caption data and 20% QA data). Finally, Stage 3 focuses on **Audio Output Tuning**, including the training of the codec model (Stage 3.1, using 3,000 hours of text-speech data) and speech decoder training (Stage 3.2). The percentages shown in the image correspond to the data sampling ratios specified in Table 1.

### 3.3.2 Stage 2: Audio Input Tuning

**Stage 2.1 Audio Alignment.** After completing the training of Stage 1, the model has developed a strong foundation in image and video understanding. In this stage, our goal is to reduce the discrepancy between audio and language based on Stage 1, enabling the LLM to understand audio inputs. The training data consists of 11,000 hours of speech-transcription pairs. We follow a two-step approach: (a) *Speech Encoder Training*: We adopt a training framework used in common speech recognition systems, using a Connectionist Temporal Classification (CTC) loss function [54] to train the speech encoder. The aim is for the encoder to predict the transcription text from the speech input. This step ensures that the audio encoder can extract speech features and map them to the text representation space. (b) *Speech Adapter Training*: After training the speech encoder, we integrate it with the LLM, using an audio adapter to introduce audio features into the input layer of the LLM. The training objective at this stage is to enable the LLM to output the transcription text of the speech data.

Besides, in step (b), we introduce special trainable input tokens to guide the speech understanding process. These tokens provide additional contextual information that guides the LLM used for the QA task to perform the ASR task.

**Stage 2.2 Audio SFT.** The focus of this stage is to introduce the QA functionality with speech questions and text answers. To achieve this, we sample 4% of the caption data and 20% of the QA data from Table 1. In terms of data processing, approximately half of the text-based questions are randomly replaced with their corresponding speech versions, generated using a TTS system.

In this stage, both the visual encoder and adapter, the audio encoder and adapter, as well as the LLM are trainable, aiming to improve the model's adaptability with multimodal inputs. In addition, we add a classification head to the LLM's output. This head is used to distinguish whether the input comes from speech or text. As a result, the model can more accurately interpret speech inputs and process different modalities efficiently and flexibly.

### 3.3.3 Stage 3: Audio Output Tuning

In the first two stages of training, the VITA-1.5 model has effectively developed its multimodal understanding capabilities. However, a crucial capacity, i.e., speech output, remains absent, which is essential for its role as an interactive assistant. To introduce speech output functionality without compromising the model's fundamental abilities, we draw on the strategy [40], using 3,000 hours of text-speech data and employing a two-step training approach (see Fig. 3).

**Stage 3.1 Codec Training.** The goal of this step is to train a codec model with a single codebook using speech data. The encoder of the codec model has the ability to map speech to discrete tokens, while the decoder can map the discrete tokens back to speech stream. During the inference phase of VITA-1.5, only the decoder is used.

**Stage 3.2 NAR + AR Decoder Training.** The training of this stage uses text-speech paired data, where the text is fed into the tokenizer and the embedding later of the LLM to obtain its embedding vectors, and the speech is fed into the encoder of the codec model to obtain its speech tokens. The text embedding vectors are sent to the NAR speech decoder to get global semantic features, and then the features are sent to the AR speech decoder, which predicts the corresponding speech tokens. Note that the LLM is frozen during this stage, thus the multimodal performance is not affected.

## 4 Evaluation

### 4.1 Vision-Language Evaluation

**Baselines.** We compare a series of open-source MLLMs, including VILA-1.5 [55], LLaVA-Next [56], CogVLM2 [57], InternLM-XComposer2.5 [58], Cambrian-1 [24], MiniCPM-V-2.6 [26], Ovis1.5 [59], InternVL-Chat-1.5, InternVL-2 [60], LLaVA-OV [47], and Video-LLaVA [30], SliME [28], and LongVA [29], as well as 5 closed-source MLLMs, including GPT-4V[7], GPT-4o[8], GPT-4o-mini, Gemini 1.5 Pro [38], and Claude 3.5 Sonnet[9].

---

[7] https://openai.com/index/gpt-4v-system-card/

[8] https://openai.com/index/hello-gpt-4o/

[9] https://www.anthropic.com/news/claude-3-5-sonnet

Table 2: **Evaluation on Image Understanding Benchmarks.** VITA-1.5 shows performance comparable to the leading open-source models and advanced closed-source counterparts. MMB refers to MMBench, MMS to MMStar, Hal to HallusionBench, MathV to MathVista, and OCR to OCRBench. Note that after the training of Stages 2 (Audio Input Tuning) and 3 (Audio Output Tuning), VITA-1.5 retains almost its original visual-language capabilities in Stage 1 (Vision-Language Training).

| Method | LLM | MMB | MMS | MMMU | MathV | Hal | AI2D | OCR | MMVet | MME | Avg |
|---|---|---|---|---|---|---|---|---|---|---|---|
| VILA-1.5 | Vicuna-v1.5-13B | 68.5 | 44.2 | 41.1 | 42.5 | 39.3 | 69.9 | 460.0 | 45.0 | 1718.2 | 52.1 |
| LLaVA-Next | Yi-34b | 77.8 | 51.6 | 48.8 | 40.4 | 34.8 | 78.9 | 574.0 | 50.7 | 2006.5 | 58.3 |
| CogVLM2 | Llama3-8B-Instruct | 70.7 | 50.5 | 42.6 | 38.6 | 41.3 | 73.4 | 757.0 | 57.8 | 1869.5 | 58.8 |
| InternLM-Xcomposer2 | InternLM2-7B | 77.6 | 56.2 | 41.4 | 59.5 | 41.0 | 81.2 | 532.0 | 46.7 | 2220.4 | 61.2 |
| Cambrian | Nous-Hermes-2-Yi-34B | 77.8 | 54.2 | 50.4 | 50.3 | 41.6 | 79.5 | 591.0 | 53.2 | 2049.9 | 61.4 |
| InternVL-Chat-1.5 | InternLM2-20B | 79.7 | 57.1 | 46.8 | 54.7 | 47.4 | 80.6 | 720.0 | 55.4 | 2189.6 | 65.1 |
| Ovis1.5 | Gemma2-9B-It | 77.3 | 58.1 | 49.7 | 65.6 | 48.2 | 84.5 | 752.0 | 53.8 | 2125.2 | 66.9 |
| InternVL2 | InternLM2.5-7b | 79.4 | 61.5 | 51.2 | 58.3 | 45.0 | 83.6 | 794.0 | 54.3 | 2215.1 | 67.3 |
| MiniCPM-V 2.6 | Qwen2-7B | 78.0 | 57.5 | 49.8 | 60.6 | 48.1 | 82.1 | 852.0 | 60.0 | 2268.7 | 68.5 |
| Proprietary | | | | | | | | | | | |
| GPT-4V | - | 65.5 | 50.4 | 59.3 | 48.2 | 39.3 | 71.4 | 678.0 | 49.0 | 1790.3 | 58.5 |
| GPT-4o mini | - | 76.0 | 54.8 | 60.0 | 52.4 | 46.1 | 77.8 | 785.0 | 66.9 | 2003.4 | 66.3 |
| Gemini 1.5 Pro | - | 73.9 | 59.1 | 60.6 | 57.7 | 45.6 | 79.1 | 754.0 | 64.0 | 2110.6 | 67.2 |
| GPT-4o | - | 82.8 | 61.6 | 62.8 | 56.5 | 51.7 | 77.4 | 663.0 | 66.5 | 2328.7 | 69.3 |
| Claude3.5 Sonnet | - | 78.5 | 62.2 | 65.9 | 61.6 | 49.9 | 80.2 | 788.0 | 66.0 | 1920.0 | 69.3 |
| Open Source | | | | | | | | | | | |
| VITA-1.0 | Mixtral-8x7B | 71.8 | 46.4 | 47.3 | 44.9 | 39.7 | 73.1 | 678.0 | 41.6 | 2097.0 | 57.8 |
| VITA-1.5 (Stage 1) | Qwen2-7B | 77.1 | 59.1 | 53.1 | 66.2 | 44.1 | 80.3 | 752.0 | 51.1 | 2311.0 | 67.1 |
| VITA-1.5-Audio (Stage 3) | Qwen2-7B | 76.7 | 59.9 | 52.1 | 66.2 | 44.9 | 79.3 | 732.0 | 49.6 | 2352.0 | 66.8 |

Table 3: **Evaluation on Video Understanding Benchmarks.** Although VITA-1.5 still lags behind models like GPT-4o and Gemini-1.5-Pro, it performs comparably to many open-source models. Note that after the training of Stages 2 (Audio Input Tuning) and 3 (Audio Output Tuning), VITA-1.5 retains almost its original visual-language capabilities in Stage 1 (Vision-Language Training).

| Method | LLM | Video-MME w/o sub | Video-MME w/ sub | MVBench | TempCompass |
|---|---|---|---|---|---|
| Video-LLaVA | Vicuna-v1.5-13B | 39.9 | 41.6 | | 49.8 |
| SliME | Llama3-8B-Instruct | 45.3 | 47.2 | - | - |
| LongVA | Qwen2-7B | 52.6 | 54.3 | - | 57.0 |
| VILA-1.5 | Llama3-8B-Instruct | - | - | - | 58.8 |
| InternLM-XComposer-2.5 | InternLM2-7B | - | - | - | 62.1 |
| LLaVA-OneVision | Qwen2-7B | 58.2 | 61.5 | 56.7 | 64.2 |
| InternVL-2 | InternLM2.5-7b | - | - | - | 66.0 |
| MiniCPM-V-2.6 | Qwen2-7B | 60.9 | 63.7 | - | 66.3 |
| Proprietary | | | | | |
| GPT-4o-mini | - | 64.8 | 68.9 | - | |
| Gemini-1.5-Pro | - | 75.0 | 81.3 | - | 67.1 |
| GPT-4o | - | 71.9 | 77.2 | - | 73.8 |
| Open Source | | | | | |
| VITA-1.0 | Mixtral-8x7B | 55.8 | 59.2 | - | 62.3 |
| VITA-1.5 (Stage 1) | Qwen2-7B | 56.8 | 59.5 | 56.8 | 65.5 |
| VITA-1.5 (Stage 3) | Qwen2-7B | 56.1 | 58.7 | 55.4 | 66.7 |

**Evaluation Benchmarks.** To assess the image perception and understanding capabilities of VITA-1.5, we utilize several evaluation benchmarks, including MME [61], MMBench [62], MMStar [63], MMMU [64], MathVista [65], HallusionBench [66], AI2D [67], OCRBench [68], and MMVet [69]. These benchmarks cover a wide range of aspects, including general multimodal capabilities (e.g., MME, MMBench, and MMMU), mathematical reasoning (MathVista), hallucination detection (HallusionBench), chart (AI2D) and OCR (OCRBench) understanding, providing a comprehensive evaluation results. For video understanding, we use representative evaluation benchmarks including Video-MME [70], MVBench [71], and TempCompass [72].

**Vision-Language Capabilities.** Table 2 presents a comparison of VITA-1.5's image understanding performance. After the training of the three stages, VITA-1.5 performs comparably to the most advanced open-source models and even surpasses some closed-source models like GPT-4V and GPT-4o-mini. This result highlights the robust capabilities of VITA-1.5 in image-language tasks. As shown in Table 3, VITA-1.5 shows comparable performance to the top open-source models in the evaluation of video understanding. The notable gap compared to proprietary models suggests that VITA-1.5 still has significant room for improvement and potential for further enhancement in video understanding. Please note that after the training of Stages 2 (Audio Input Tuning) and 3 (Audio Output Tuning), VITA-1.5 retains almost its original visual-language capabilities in Stage 1 (Vision-Language Training).

Table 4: **Evaluation on ASR Benchmarks.** VITA-1.5 has demonstrated strong performance in both Mandarin and English ASR tasks. It outperforms specialized speech models, achieving better results in both languages.

| Model | CN (CER↓) | | | Eng (WER↓) | | | |
|---|---|---|---|---|---|---|---|
| | aishell-1 | test net | test meeting | dev clean | dev other | test clean | test other |
| Wav2vec2-base | - | - | - | 6.0 | 13.4 | - | - |
| Mini-Omni2 | - | - | - | 4.8 | 9.8 | 4.7 | 9.4 |
| Freeze-Omni | 2.8 | 12.6 | 14.2 | 4.2 | 10.2 | 4.1 | 10.5 |
| VITA-1.0 | - | 12.2 | 16.5 | 7.6 | 16.6 | 8.1 | 18.4 |
| VITA-1.5 | **2.2** | **8.4** | **10.0** | **3.3** | **7.2** | **3.4** | **7.5** |

## 4.2 Speech Evaluation

**Baselines.** The following three baseline models are used for comparison: Wav2vec2-base [73], Mini-Omni2 [74], Freeze-Omni [40], and VITA-1.0 [12].

**Evaluation Benchmarks.** *The Mandarin Evaluation Sets* consists of three datasets: aishell-1 [75], test net [76], and test meeting [77]. These datasets are used to evaluate the model's performance on Mandarin speech. The evaluation metric is the Character Error Rate (CER). *The English Evaluation Sets* include four datasets: dev-clean, dev-other, test-clean, and test-other [78], which are used to evaluate the model's performance on English speech. The evaluation metric is Word Error Rate (WER). The evaluation results in Table 4 indicate that VITA-1.5 achieves leading accuracy in both Mandarin and English ASR tasks. This demonstrates that VITA-1.5 has successfully integrated advanced speech capability to support multimodal interaction.

## 5 Conclusion and Future Work

In this paper, we has presented VITA-1.5, a multimodal LLM designed to integrate vision and speech through a carefully crafted three stage training strategy. By relieving the inherent conflicts between modalities, VITA-1.5 achieves robust capabilities in both vision and speech understanding, enabling efficient speech-to-speech interactions without relying on separate ASR or TTS modules. Extensive evaluations demonstrate that VITA-1.5 performs competitively across multimodal benchmarks. We hope that VITA-1.5 can promote the progress of open-source models in the field of real-time multimodal interaction. Although VITA-1.5 has made some contributions, such as multi-modality joint training, end-to-end architecture, response latency, and basic performance, there are two major areas that can be improved in our future work:

1. Personalized MLLM. Currently, VITA-1.5 is generic and do not incorporate individual preferences during interaction. For example, after learning about personal preferences in the interaction, the content and manner of answers can be adjusted accordingly.

2. Long-term memory. The process of human-computer interaction can last 10 minutes or even several hours, in which case it is important for the human-computer interaction in real scenarios.

## Impact Statement

This paper studies the technology of large models to enhance their technical level. Its influence is the same as that of the research on other large models and will not be repeated here.

## Acknowledgments

This work is funded by National Natural Science Foundation of China (Grant No. 62506158 and No. 62441234), Fundamental Research Funds for the Central Universities, AI & AI for Science Project of Nanjing University (No. 2024300529), and CCF-Tencent Rhino-Bird Open Research Fund.

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
