# OpenReview forum: "VITA-1.5: Towards GPT-4o Level Real-Time Vision and Speech Interaction"
_NeurIPS.cc/2025/Conference — NeurIPS 2025 spotlight_

### Official Review · Reviewer_RiXE · 2025-06-23

**Clarity:** 1
**Significance:** 2
**Originality:** 1
**Rating:** 2
**Confidence:** 4

**Summary:**

- The paper presents VITA-1.5, an end-to-end multmodal model capable of tackling video/audio/image understanding tasks with speech and text outputs.
- The authors introduce a comprehensive set of datasets with a multi-stage training technique to develop the proposed model.

**Questions:**

- The video shared in the appendix is in chinese, how well does it perform with english audio/speech?
- How much compute was used to train the model?

**Ethical Concerns:**

["NO or VERY MINOR ethics concerns only"]

**Final Justification:**

During the rebuttal and discussion phase, the authors shared a few more details about their work, however, all the useful details are missing from the current article and i feel it is not right to accept it in the current state. It requires many more ablations and dataset curation details/experiments to have publishable quality.

**Limitations:**

Yes

**Paper Formatting Concerns:**

The authors list VITA-1.0 under "Ours" in Tab. 2. If true, it makes their identity known because the paper is cited on line 103, if false, that is very misleading and a big blunder.

**Quality:**

1

**Strengths And Weaknesses:**

### Strength

- The model is able to output speech and is end-to-end.

### Weaknesses

- I have trouble finding a technical takeaways from this paper that could help research move forward. There are no details shared about data curation, the model architecture follows VITA-1.0 to most extent, no real ablations on different training stages or dataset choices.
- The authors say in the checklist that they do not provide access to data stating that its widely available with preprocessing details shared but i do not find the mention of that for any kind of data. They are very vague about the audio and speech data used in sec 3.3.2 and 3.3.3, so that justification is wrong.
- Do the authors plan to release details or checkpoints?
- The results lack behind most methods in Tab. 2

---

> ### Author Rebuttal · Authors · 2025-07-30
>
> Sincerely thanks for your efforts in reviewing this work. We hope the detailed responses help clarify your concerns. We would greatly appreciate it if you could kindly re-evaluate our work in light of the new explanations and additional results.
>
> ------
>
> **Q1: I have trouble finding a technical takeaways from this paper that could help research move forward. There are no details shared about data curation, the model architecture follows VITA-1.0 to most extent, no real ablations on different training stages or dataset choices.**
>
> **R1**: We respectfully clarify that our paper presents several important technical takeaways that we believe are valuable for advancing multimodal interaction systems:
>
> 1. **Progressive Multi-Stage Curriculum to Alleviate Modality Interference**
>    While VITA-1.0 proposed an initial coarse-grained multi-stage pipeline, VITA-1.5 introduces a significantly refined **six-phase training strategy**. This curriculum gradually incorporates different modalities in a stabilized order—first vision, then audio input, and finally audio output.
>    This progressive approach **mitigates modality interference** and maintains the core linguistic capabilities of the base LLM (e.g., <1% variation on MMLU and GPQA-Diamond), which we believe is a **methodologically valuable insight** for future MLLM training paradigms.
>
> 2. **End-to-End Real-Time Speech Interaction**
>    VITA-1.5 upgrades from the cascaded TTS pipeline in VITA-1.0 to an **end-to-end speech generation system** using codec tokens.
>    This architectural shift enables **low-latency speech output** (↓1.4s) for nearly real-time interaction—an important system-level contribution to interactive AI agents.
>
>
> 3. **Data Handling and Reproducibility**
>    The training data used in this work are from **publicly available datasets**, including audio (internally organized open-source audio data), vision, and multimodal instruction corpora. Due to space constraints, we briefly described audio data use in Sec 3.3.2 and 3.3.3, but we acknowledge the descriptions were insufficiently detailed.
>    In the revised version, we will:
>    - Provide a **complete list of datasets**,
>    - Clarify **data preprocessing and filtering procedures**, and
>    - **Release training&inference codes and model checkpoints** to ensure **reproducibility**.
>
> 4. **Ablation Studies**
>    We agree that ablations would strengthen the contribution. Due to time limitations, we prioritized delivering an integrated, end-to-end functioning system. We are currently supplementing ablations on training order and data mixing ratios, and plan to include them in final version.
>
> We will make these contributions and distinctions more explicit in the revised manuscript.
>
> ------
>
> **Q2: The authors say in the checklist that they do not provide access to data stating that its widely available with preprocessing details shared but i do not find the mention of that for any kind of data. They are very vague about the audio and speech data used in sec 3.3.2 and 3.3.3, so that justification is wrong.**
>
> **R2**: We acknowledge that Sections 3.3.2 and 3.3.3 lack sufficient detail regarding the audio and speech datasets used, and we appreciate the opportunity to clarify and improve this aspect.
>
> The data used in this work are sourced from **publicly available datasets**. Specifically:
>
> - For **audio input (ASR)**, we use well-established datasets such as **AISHELL-1**, **Common Voice**, and **LibriSpeech**.
> - For **speech output (TTS)**, open-source text corpora are translated to audio by a TTS system.
> - For **instruction tuning**, we synthesize multimodal instruction-style annotations based on these datasets to align with the model’s multi-turn conversational capabilities.
>
> To ensure **reproducibility**, we will:
> - Provide a **list of used datasets** in the final version, including download links;
> - Describe our **data preprocessing pipeline** in detail (e.g., how to use TTS systems to synthesize audios);
> - **Release our training&inference codes and model checkpoints**.
>
> We are fully committed to open research and making VITA-1.5 reproducible, including all necessary resources for the community to replicate our results and build upon our work.
>
> ------
>
> **Q3: Do the authors plan to release details or checkpoints?**
>
> **R3**: Yes — we will **definitely release** the training&inference codes and model checkpoints. Ensuring **reproducibility and openness** is a core goal of our work. We will also provide clear documentation and usage instructions to support the community.
>
> We are fully committed to supporting open research and lowering the barrier for future work in multimodal interaction.
>
> ------
>
> **Q4: The results lack behind most methods in Tab. 2.**
>
> **R4**: We appreciate the reviewer’s concern regarding performance in Table 2. While it is true that VITA-1.5 slightly lags behind certain **closed-source models** such as GPT-4o (66.8 vs. 69.3), we would like to emphasize several important clarifications:
>
> 1. **Strong Performance Compared to Open Models**
>    VITA-1.5 achieves highly competitive performance compared to advanced **open-source models**. For instance, it is only **0.5 points behind InternVL2** (66.8 vs. 67.3) in average performance, despite InternVL2 using many private training data and focusing solely on image-text tasks without supporting audio or speech interaction.
>
> 2. **Multimodal Interaction Capability**
>    Unlike many vision-language models such as InternVL2, VITA-1.5 supports **end-to-end multimodal interaction**, including nearly real-time **speech input/output** and **speech interruption handling**, making it more broadly applicable to real-world interactive scenarios.
>
> 3. **Competitive Results vs. Fully Open Models**
>    Compared to **fully open-source baselines**, such as LLaVA-OneVision, which also use only publicly available training data, VITA-1.5 demonstrates **consistently stronger performance** across benchmarks:
>
>    | Model                 | MME  | MMMU | MMBench | MathVista | MMStar |
>    |----------------------|------|------|---------|-----------|--------|
>    | LLaVA-OneVision-7B   | 1998 | 48.8 | 80.8    | 63.2      | 61.9   |
>    | **VITA-1.5-7B**       | **2311** | **52.1** | 76.6    | **66.2**  | 60.2   |
>
>    These results highlight the **solid multimodal foundation** of VITA-1.5, achieved while simultaneously supporting speech modalities — which LLaVA-OneVision does not.
>
> In summary, VITA-1.5 strikes a meaningful balance between **broad modality coverage** and **strong open-source benchmark performance**, and we believe this makes it a valuable step forward in the direction of real-time, speech-enabled MLLMs.
>
> ------
>
> **Q5: The video shared in the appendix is in chinese, how well does it perform with english audio/speech?**
>
> **R5**: While the demonstration video in the appendix is in Chinese, we confirm that **VITA-1.5 fully supports both Chinese and English** for speech input and output. In fact, we evaluate English speech performance using standard benchmarks. For example, in Table 4, VITA-1.5 achieves better than Wav2vec2-base, Mini-Omni2, and Freeze-Omni on the English ASR tasks. The English WER on the Seed-TTS-eval benchmark is 8.44% (use only 3K hours training data), which has demonstrated the TTS performance.
>
> We will provide an English version of the demo video in the final submission to better showcase these capabilities.
>
> ------
>
> **Q6: How much compute was used to train the model?**
>
> **R6**: The training took 5,300+ H20 GPU hours.

---

> ### Comment · Reviewer_RiXE · 2025-08-03
>
> Thank you to the authors for the rebuttal. I suggest they incorporate the mention details in the future revisions along with ablations covering all the dataset decisions that they have described.
>
> This model took 5300+ H20 GPU hours to train, rendering the findings from this paper inaccessible to labs with less resources. The authors should consider training a small scale academic model. That would be more useful to the community to decide if such a model can even be developed with less data and resources and would be easier to reproduce.
>
> Also, VITA-1.5-7B does not perform on par with llava-ov-7b on mmstar which is the only "good" benchmark nowadays, other benchmarks shared by the authors have well-known flaws, making it not too useful for image reasoning.

---

> ### Author Response · Authors · 2025-08-03
> **Response to the new concerns**
>
> We sincerely thank you for the follow-up comments. Below we address the new concerns.
>
> ------
>
> **Q1: Comment on compute cost and accessibility.**
>
> **R1**:
> We recognize the need for lightweight academic versions. In future work, we plan to train and release smaller variants (e.g., 2B) for research reproducibility under limited resources.
>
> We believe that **releasing training&inference codes, checkpoints, and clear recipes from a 7B model still provides substantial value to the community**. The **7B scale is currently a widely adopted and practical size** in open-source research, striking a good balance between capability and accessibility.
>
> ------
>
> **Q2: Concern that “the benchmarks have well-known flaws’’**.
>
> **R2**: We respectfully clarify that, with the exception of MMStar, the other benchmarks we use—including **MME**, **MMMU**, **MMBench**, and **MathVista**—are **widely adopted by many representative models** in both the closed-source and open-source communities. For example:
> * **GPT-4.1** adopts  **MMMU** and **MathVista** as primary evaluation benchmarks.
> * **Qwen2.5-VL** and **InternVL-3** report results on **MME**, **MMBench**, **MMMU**, and **MathVista**.
>
> These benchmarks have become de facto standards for evaluating **multimodal understanding and reasoning** across the community, including for new model releases from top-tier institutions and companies.
> **VITA-1.5-7B outperforms LLaVA-OneVision-7B** on **MME**, **MMMU**, and **MathVista**, demonstrating its competitive vision-language capabilities:
>
> | Model | MME | MMMU | MathVista |
> | ------------------ | -------- | -------- | --------- |
> | LLaVA-OneVision-7B | 1998 | 48.8 | 63.2 |
> | **VITA-1.5-7B** | **2311** | **52.1** | **66.2** |
>
> We fully agree that **all benchmarks have some inherent limitations**. However, the widespread adoption of these tests **reflects their practical utility and community recognition to some extent**.
>
> Furthermore, **to address concerns about benchmark robustness**, we have also evaluated our model on **MMMU-Pro** (10 Options) , an upgraded version of MMMU that resolves several issues proposed by MMStar.
> On **MMMU-Pro**, **VITA-1.5-7B again outperforms LLaVA-OneVision-7B**:
>
> | Model | MMMU-Pro|
> | ------------------ | --------------------- |
> | LLaVA-OneVision-7B | 29.7 |
> | **VITA-1.5-7B** | **30.4** |
>
> This result further confirms the strong image reasoning ability of VITA-1.5 under more rigorous evaluation. Besides, VITA-1.5 also supports the **speech modality**, which is **not available in LLaVA-OneVision**.
>
> In conclusion, while we acknowledge that no benchmark is perfect, our use of **well-accepted, widely-used evaluations**, and the inclusion of newer benchmarks like **MMMU-Pro**, provides a solid and fair basis for assessing VITA-1.5's performance and progress in multimodal understanding and reasoning.

---

> ### Comment · Reviewer_RiXE · 2025-08-06
>
> Thank  you to the authors for the reply!
>
> The gains on MMMU-Pro are impressive. However, I still believe this work is still a WIP in terms of how much value it can contribute to the community. The 5300+ H20 hours are not only due to the 7B model but also due to the amount of data used by the authors, details about which are still not fully known. The authors say they will release the full data, however, it is next to impossible for everyone to train even a 2B model with so much (could the authors clarify the total number of samples?) data. I believe there is a lot of promise in this paper but the current draft and experiments are not enough to advance the community in terms of developing better systems. With a lot of compute at the authors' disposal, I hope to see extensive ablations and experiments about how different kinds of data affect the system -> for example:
>
> - how does the speech data affect the vision reasoning performance and vice-versa? -> is there a positive/negative or no transfer at all?
> - what kind of data helps the performance the most? -> for example, it is well known in the vision community that academic QnAs help performance on benchmarks a lot but not on general use cases, is it the same for speech too?
> - does VITA-1.5 work on videos too?
>
> in conclusion, although the other reviewers feel it's a good system, in my opinion, as a research paper/study, there's a still a lot of room for improvement to help the community and therefore, I will maintain my score.

---

> > ### Author Response · Authors · 2025-08-06
> > **Response to Follow-up Comments**
> >
> > Thanks for your follow-up. Please find our clarifications below:
> >
> > ---
> >
> > ### **1. Training Cost and Community Accessibility**
> >
> > In the large model era, **training from scratch on full data is extremely resource-intensive and not expected of most research groups**. Even for widely used models such as **Qwen-VL** or **InternVL**, few in the community retrain them fully. Instead, it is common practice to build upon open-sourced baselines through fine-tuning or task adaptation.
> >
> > To support this, we will **open-source our training/inference codes, checkpoints, and data**, making it easy for others to reproduce our results or extend the model with their own data under limited resources. We believe this is meaningful for the research community, especially for multimodal interaction.
> >
> > Besides, as described in the paper, our total data includes: ~22.1M multimodal instruction samples, ~110K hours of ASR data, and ~3K hours of TTS data.
> >
> > ---
> >
> > ### **2. On the Contributions of VITA-1.5**
> >
> > We would like to take this opportunity to briefly reiterate the core contributions of VITA-1.5:
> >
> > - The progressive, multi-stage training strategy designed to reduce modality interference. By this manner, the adding of speech has little effect on other multi-modal performance (vision-language). For example, the average image understanding performance has decreased by **no more than 0.5%**, and the model maintains the core language capabilities of the base LLM (e.g., **<1% variation** on MMLU and GPQA-Diamond).
> >
> > - The integration of end-to-end speech generation into multimodal interaction, reducing latency from ~4s to 1.5s. This makes it a nearly real-time interactive system, effectively **promoting the development of open-source multimodal interaction**.
> >
> > - Strong performance across image, video, and speech benchmarks. We will open-source our training/inference codes, checkpoints, and data, to help other members of the community continue to develop and explore multimodal interaction.
> >
> > We believe these contributions provide a **practical and extensible foundation** for future multimodal interaction research.
> >
> > ---
> >
> > ### **3. Cross-Modal Effects: Speech vs Vision**
> >
> > We have analyzed this in the paper. After adding the speech capability, the average image understanding performance has decreased by **no more than 0.5%**, validating our progressive training strategy. This insight is valuable for future MLLMs aiming to support diverse modalities without sacrificing core capabilities.
> >
> > ---
> >
> > ### **4. On Dataset Impact and Ablation**
> >
> > We agree that fine-grained data ablation would be valuable. However, in the context of large model training, this is **very difficult due to the extreme compute demand**. Our current data design is based on prior experience, and during training, we continuously monitored core metrics such as vision-language performance and base LLM ability to ensure stability and effectiveness.
> >
> > ---
> >
> > ### **5. Video Support**
> >
> > Yes, VITA-1.5 supports **video input**, and the paper includes results on **MVBench**, **TempCompass**, and **Video-MME**. VITA-1.5 outperforms some open-source video MLLM such as LongVA.
> >
> > ---
> >
> > ### **Conclusion**
> >
> > While we acknowledge there is always room for future analysis and improvement, we believe this work provides a **valuable and practical contribution** to the community by delivering **a strong open baseline for nearly real-time multimodal interaction**, with full support for reproducibility and extension.

---

### Official Review · Reviewer_RHor · 2025-06-30

**Clarity:** 3
**Significance:** 3
**Originality:** 2
**Rating:** 5
**Confidence:** 4

**Summary:**

This paper introduces an omni-modal model, VITA-1.5. The training of VITA consists of three stages: the first stage focuses on acquiring visual understanding, the second stage develops speech understanding, and the third stage enables speech generation. VITA-1.5 achieves strong performance across image, video, and speech benchmarks. Demonstrations show that VITA-1.5 is capable of real-time interaction and supports features such as speech interruption.

**Questions:**

1. **Speech Interruption.** The video demonstration showcases the model’s ability to handle speech interruption. The authors are encouraged to include a description in the paper detailing how this feature is implemented, including what modules were introduced and which data were used.

2. **Speech decoder.** The speech decoder in this paper consists of 4 layers of AR decoder and 4 layers of NAR decoder. The authors should clarify the motivation for this design choice, rather than using only AR or only NAR decoders. It would be helpful if the authors could provide an explanation or include ablation studies to justify this architectural decision.

**Ethical Concerns:**

["NO or VERY MINOR ethics concerns only"]

**Final Justification:**

The author's detailed rebuttal has basically addressed my concerns. After reading the author's responses to the other reviewers' comments, I have decided to raise my score to 5.

**Limitations:**

yes

**Quality:**

3

**Strengths And Weaknesses:**

### Strengths

1. **Paper writing.** This paper is well written and easy to follow; the design of the 3-stage training pipeline is reasonable.

2. **Interesting demo.** This paper provides a demo video that supports speech interruption and works in real-time. I like this demo video.

### Weakness

1. **Streaming speech output.** The speech decoder consists of 4 layers of NAR decoder and 4 layers of AR decoder. Does the NAR mean that full attention, rather than causal attention, is conducted across all audio tokens? If yes, then I think VITA-1.5 is hard to support streaming speech output. Users have to wait until all the text is generated before they can hear the speech output.

2. **Lack of comparison.** VITA-1.5 achieves good performance on various image, video and speech benchmarks. The author should add comparisons with some new open-source MLLMs, such as LLaVA-Onevision[1], and Qwen2-VL[2] (It is okay if the performance is lower than Qwen2-VL series, since they didn't release training data).

3. **Technical novelty.** This paper introduces a three-stage pipeline for training an Omni Model. However, the training pipeline and model architecture described are quite similar to those of other models in the field, such as Mini-Omni2[3], Llama-Omni[4], and Lyra[5]. As a research paper, I think it is important for the authors to clearly highlight the methodological novelty of VITA-1.5, compared to other related works.

[1] Li, Bo, et al. "Llava-onevision: Easy visual task transfer." arXiv preprint arXiv:2408.03326 (2024).

[2] Wang, Peng, et al. "Qwen2-vl: Enhancing vision-language model's perception of the world at any resolution." arXiv preprint arXiv:2409.12191 (2024).

[3] Xie, Zhifei, and Changqiao Wu. "Mini-omni2: Towards open-source gpt-4o with vision, speech and duplex capabilities." arXiv preprint arXiv:2410.11190 (2024).

[4] Fang, Qingkai, et al. "Llama-omni: Seamless speech interaction with large language models." arXiv preprint arXiv:2409.06666 (2024).

[5] Zhong, Zhisheng, et al. "Lyra: An Efficient and Speech-Centric Framework for Omni-Cognition." arXiv preprint arXiv:2412.09501 (2024).

---

> ### Author Rebuttal · Authors · 2025-07-30
>
> We sincerely thanks for your positive recognition of our work. Our point-by-point responses to all your comments are provided below.
>
> ------
>
> **Q1: Streaming speech output. The speech decoder consists of 4 layers of NAR decoder and 4 layers of AR decoder. Does the NAR mean that full attention, rather than causal attention, is conducted across all audio tokens? If yes, then I think VITA-1.5 is hard to support streaming speech output. Users have to wait until all the text is generated before they can hear the speech output.**
>
> **R1**: Thank you for your insightful question. Indeed, the NAR decoder in VITA-1.5 applies full attention across input tokens. However, VITA-1.5 is specifically designed to support **low-latency, streaming speech output** through a hybrid decoding strategy:
>
> - The NAR decoder operates in a **sentence-by-sentence manner**, where sentences are segmented by punctuation (e.g., commas). These segments are typically short, allowing efficient chunk-wise speech token generation without requiring the full text beforehand.
> - The AR decoder follows and performs causal decoding over the generated tokens, enabling **real-time, progressive speech synthesis**.
> - Most importantly, the **generation rate of text tokens is significantly faster than that of speech tokens**. On NVIDIA H20 GPUs, generating speech tokens corresponding to 1 second of text typically takes 3.2 seconds. Additionally, waveform decoding and audio playback introduce further delay.
>
> This natural gap means that **text generation typically stays ahead of speech synthesis**. Thus, the chunk-wise NAR decoding **does not become a bottleneck** for streaming, as long as the first sentence is short, allowing the system to start generating and playing speech almost immediately. **Users do not have to wait until the full text is generated** to begin hearing the response.
>
> In summary, while the NAR decoder is non-causal, VITA-1.5 effectively supports streaming speech output in practice through architectural design and scheduling strategies. We will make this explanation more explicit in the revised version.
>
> ------
>
> **Q2: Lack of comparison. VITA-1.5 achieves good performance on various image, video and speech benchmarks. The author should add comparisons with some new open-source MLLMs, such as LLaVA-Onevision[1], and Qwen2-VL[2] (It is okay if the performance is lower than Qwen2-VL series, since they didn't release training data).**
>
> **R2**: Thanks for your valuable suggestion. We have conducted additional comparisons between VITA-1.5 and recent open-source multimodal models, specifically **Qwen2-VL-7B** and **LLaVA-OneVision-7B**, on widely adopted benchmarks including **MME**, **MMMU**, **MMBench**, **MathVista**, and **MMStar**. The results are summarized below:
>
> | Model                | MME  | MMMU | MMBench | MathVista | MMStar |
> |----------------------|------|------|---------|-----------|--------|
> | **Qwen2-VL-7B**        | 2276 | 53.7 | 81.0    | 61.6      | 60.7   |
> | **LLaVA-OneVision-7B** | 1998 | 48.8 | 80.8    | 63.2      | 61.9   |
> | **VITA-1.5 (ours)**    | 2311 | 52.1 | 76.6    | 66.2  | 60.2   |
>
> As shown:
> - VITA-1.5 achieves the **highest score on MME and MathVista**, demonstrating strong **perception and reasoning** capabilities, especially in math-related tasks;
> - VITA-1.5 performs competitively on other benchmarks, despite its additional support for **audio and speech interaction**, which are not covered by the above baselines;
> - We emphasize that VITA-1.5 is designed as an **omni** model with **real-time speech interaction**, whereas Qwen2-VL and LLaVA-OneVision focus solely on **vision-language** tasks.
>
> We will include these comparison results in the revised version.
>
> ------
>
> **Q3: Technical novelty. This paper introduces a three-stage pipeline for training an Omni Model. However, the training pipeline and model architecture described are quite similar to those of other models in the field, such as Mini-Omni2[3], Llama-Omni[4], and Lyra[5]. As a research paper, I think it is important for the authors to clearly highlight the methodological novelty of VITA-1.5, compared to other related works.**
>
> **R3**: Thanks for the thoughtful comment on the novelty of our approach. We would like to emphasize that while VITA-1.5 shares high-level goals with recent works such as Mini-Omni2, LLaMA-Omni, and Lyra (**will all be cited in our revised version**), our contributions lie in the **systematic, scalable, and practical realization** of real-time omni-modal interaction. Specifically, VITA-1.5 distinguishes itself from these works in the following key aspects:
>
> 1. **Systematic Three-Stage Progressive Training Pipeline**
> While these works often adopt a one-shot or loosely staged training approach, VITA-1.5 introduces a **carefully crafted three-stage curriculum** that incrementally incorporates vision and speech modalities while alleviating cross-modal conflicts. This includes:
> - Stage I: Vision-language alignment and instruction tuning.
> - Stage II: Audio input integration with fine-grained CTC alignment and modality-type classification.
> - Stage III: End-to-end speech generation via codec + NAR+AR decoding.
> This structure ensures **stable convergence and modular controllability**, which is not present in Mini-Omni2 or LLaMA-Omni.
>
> 2. **Sentence-wise NAR + AR Speech Decoder Design for Streaming Output**
> VITA-1.5 proposes a **sentence-segmented NAR** followed by an **AR decoder for causal streaming**, enabling both fast generation and real-time playback. Unlike Lyra, which focuses on efficiency, our method **balances latency and fidelity** with a decoding design tailored for **interactive speech responses**.
>
> 3. **Unified and Practical Real-Time System Deployment**
> VITA-1.5 has been **deployed in a real-world demo**, achieving <700ms model latency (on A800), and ~1.5s total latency (including preprocessing and audio playback). To our knowledge, it is the **first open omni-modal system** that demonstrates this level of **real-time vision and speech interaction** in a single end-to-end pipeline.
>
> 4. **Robust Benchmark Results Across Modalities**
> Our model achieves **competitive results** on vision-language tasks (e.g., MathVista, MME), while also delivering strong **speech recognition performance**. This demonstrates that **adding speech capability has little impact on vision performance**, which is a known challenge in multimodal joint training.
>
> In summary, the **novelty of VITA-1.5 lies not in isolated architectural tweaks**, but in its **coherent, practical, and extensible framework** that effectively brings together high-quality perception, reasoning, and real-time interaction across modalities. We will clarify and highlight these novel aspects more explicitly in the revised version.
>
> ------
>
> **Q4: Speech Interruption. The video demonstration showcases the model’s ability to handle speech interruption. The authors are encouraged to include a description in the paper detailing how this feature is implemented, including what modules were introduced and which data were used.**
>
> **R4**: Thank you for your insightful question. In VITA-1.5, **speech interruption** is supported through a **duplex inference design**, which deploys **two identical models** in parallel and coordinates their responses via a lightweight voice activity detection (VAD) mechanism.
>
> Specifically:
> - During duplex interaction, both **Model-A** and **Model-B** are initialized identically.
> - When the user issues **Query-1**, Model-A starts generating the spoken response.
> - If the user begins **Query-2** while Model-A is still speaking, the **VAD detects user speech onset**, and the system **immediately stops Model-A’s decoding**.
> - At the same time, **Model-B takes over** and starts generating a response to Query-2, ensuring smooth and low-latency interaction.
>
> This design offers several advantages:
> - It allows **true real-time bidirectional control**, as models can be switched instantly without reloading or context corruption.
> - The interruption logic is implemented **at the orchestration level**, without modifying the model architecture.
> - The use of **autoregressive speech decoding** enables token-level control and precise halting of generation.
>
> While we did not use explicitly labeled "interruption" data during training, VITA-1.5 was trained on **dialog-style multimodal instruction tuning data**, which implicitly encourages responsiveness and turn-taking behavior.
>
> We will update the paper to explain more clearly.
>
> ------
>
> **Q5: Speech decoder. The speech decoder in this paper consists of 4 layers of AR decoder and 4 layers of NAR decoder. The authors should clarify the motivation for this design choice, rather than using only AR or only NAR decoders.**
>
> **R5**: Thank you for raising this important point. The design of the VITA-1.5 speech decoder is motivated by the need to **balance latency, fluency, and controllability** in speech generation.
>
> - **NAR decoder (fast and global)**: The NAR decoder operates in parallel and produces a **coarse global layout** of speech tokens conditioned on the semantic content of the text. This enables **low-latency initial speech token generation** and facilitates **sentence-level semantic alignment**.
> - **AR decoder (fine and causal)**: The AR decoder then performs **causal refinement** of the NAR output, generating high-quality speech tokens **token-by-token**. This step ensures **natural prosody, smoothness**, and compatibility with **streaming playback**.
>
> Otherwise:
> - Using **only AR decoding** would result in better quality but **significantly increases latency**, especially for long responses, which hurts real-time user experience.
> - Using **only NAR decoding** would reduce latency but often leads to **unnatural prosody and token artifacts**, particularly in expressive or long-form speech.
>
> We will include the explanation in the revised version.

---

> > ### Comment · Reviewer_RHor · 2025-08-05
> >
> > The author's detailed rebuttal has basically addressed my concerns. After reading the author's responses to the other reviewers' comments, I have decided to raise my score to 5.

---

> > > ### Author Response · Authors · 2025-08-05
> > >
> > > We sincerely appreciate your positive feedback.  We are grateful for your recognition and the raised score.

---

### Official Review · Reviewer_Zxsq · 2025-07-02

**Clarity:** 2
**Significance:** 2
**Originality:** 2
**Rating:** 4
**Confidence:** 5

**Summary:**

VITA-1.5 is a Multimodal Large Language Model (MLLM) integrating vision, language, and speech for real-time interaction. It employs a three-stage training approach to mitigate modality conflicts: first building vision-language capabilities, then tuning for audio input (ASR), and finally training an end-to-end speech decoder (TTS). VITA-1.5 reduces interaction latency from ~4s to ~1.5s, improves performance on multimodal benchmarks, and offers strong speech processing, making it a competitive open-source model, validated across image, video, and speech benchmarks.

**Questions:**

1. Can author provide a more detailed rationale or ablation studies for the specific ordering of three-stage training pipeline?
2. What is the impact on the language model itself after each stage of training?

**Ethical Concerns:**

["NO or VERY MINOR ethics concerns only"]

**Final Justification:**

Although the quality of the generated speech seems a bit poor (WER seems high), considering the comprehensiveness of the model, I am willing to increase my score.

**Limitations:**

Yes, the authors have included a section on future work that acknowledges limitations such as the lack of personalization and long-term memory.

**Paper Formatting Concerns:**

No major formatting issues in this paper

**Quality:**

2

**Strengths And Weaknesses:**

### Strengths:
1. The paper demonstrates an impressive engineering effort in integrating multiple modalities (image, video, speech, text) and training on a massive and diverse collection of datasets. The resulting model, VITA-1.5, shows strong performance on various open-source benchmarks, often outperforming other open-source models in vision-language and ASR tasks.
2. The goal of achieving low-latency, real-time multimodal interaction with end-to-end speech I/O is highly relevant and challenging. Moving away from cascaded ASR-LLM-TTS systems is a valuable research direction to reduce error propagation and latency, and the paper makes a commendable attempt in this direction.

### Weaknesses
1. The title "Towards GPT-4o Level" sets a very high expectation that is not substantiated by the paper's own results.
2. A primary claim is the model's ability to enable "fluent speech-to-speech dialogue." While the paper provides extensive evaluation for Automatic Speech Recognition (ASR) performance (Table 4), there is a complete absence of evaluation for the Text-to-Speech (TTS) output. For an interactive model, the quality of the generated speech is paramount. There are no objective metrics (e.g., Mean Opinion Score - MOS) or subjective user studies to assess the naturalness, prosody, intonation, or speaker similarity of the generated audio. The authors claim to overcome limitations of traditional TTS systems, such as the loss of paralinguistic features, but provide no evidence to support this.
3. The three-stage training pipeline is presented as a fixed recipe. To strengthen the contribution, the authors should have provided ablations to justify their design choices. For instance, what happens if the training order is changed (e.g., Audio Input -> Vision -> Audio Output)? How sensitive is the model to the data mixing ratios described? A deeper analysis of the modality conflicts they aim to relieve would be more insightful than just showing the final successful outcome.
4. VITA 1.0 proposed a multi-stage training method, and VITA 1.5 does not seem to have made any major changes, so it is difficult to be considered a core contribution.
5. The audio input is handled by a speech encoder that processes Mel-spectrograms and maps them into the LLM's semantic space. In contrast, the audio output relies on a separate codec to generate discrete audio tokens, which the LLM is trained to predict. These input and output pathways operate in fundamentally different representation spaces. The paper does not discuss the implications of this design choice. It is unclear how this disconnect affects the model's ability to learn a unified audio representation, potentially limiting capabilities like voice cloning or maintaining prosodic consistency, and whether it introduces training inefficiencies.

---

> ### Author Rebuttal · Authors · 2025-07-30
>
> Sincerely thanks for your efforts in reviewing this work. We hope the detailed responses help clarify your concerns. We would greatly appreciate it if you could kindly re-evaluate our work in light of the new explanations and additional results.
>
> ------
>
> **Q1: The title "Towards GPT-4o Level" sets a very high expectation that is not substantiated by the paper's own results.**
>
> **R1**: Thank you for pointing this out. We agree that the title “Towards GPT-4o Level” may set high expectations, and we would like to clarify that it is intended to convey our **directional motivation**, rather than a claim of parity with GPT-4o.
>
> Specifically:
> - Our work was directly inspired by OpenAI’s **interactive demo of GPT-4o**, which showcased real-time multimodal interaction capabilities. This motivated us to explore similar **speech–vision–language integration** within the open-source community.
> - VITA-1.5 represents one of the **earliest open-source attempts** to build an **interactive, real-time multimodal LLM** with unified **speech input and output** capabilities.
> - While our model does not yet match GPT-4o in all dimensions, we believe our **design choices, training strategies, and system deployment** offer a solid step **towards that goal**, especially considering the challenges of **open data, limited compute, and reproducibility**.
>
> We will revise the paper to clarify this point and avoid any unintended overstatement in the final version.
>
> ------
>
> **Q2: Comment on lack of evaluation for TTS output.**
>
> **R2**: Thank you for pointing this out. We have conducted both **objective** and **subjective** evaluations to assess the performance of the speech generation module.
>
> 1. **Objective Evaluation**
> We evaluate our speech decoder on the **Seed-TTS-eval** benchmark using **WER** (lower is better) as a proxy for intelligibility:
> - VITA-1.5 achieves **2.63% (zh)** and **8.44% (en)** WER on test sets.
> - For comparison, CosyVoice achieves **4.08% / 4.07%**, but uses over **160K hours** of training data, while VITA-1.5 uses only **3K hours**.
> - This demonstrates that VITA-1.5 achieves competitive speech output quality even with limited data.
>
> 2. **Subjective User Study**
> To assess real-world speech quality and user experience, we conducted a **user study** involving **15 participants** who engaged in open-ended multimodal conversations (video + speech) with both VITA-1.0 and VITA-1.5. Participants rated system performance (0–5 scale) on three dimensions, including Answer Satisfaction (**AS**), Interaction Fluency (**IF**), and Interruption Naturalness (**IN**):
>
> | Model     | AS | IF | IN |
> |-----------|---------------------------|----------------------------|-------------------------------|
> | VITA-1.0  | 3.7                       | 3.5                        | 4.2                           |
> | VITA-1.5  | **4.1**                   | **4.4**                    | **4.4**                       |
>
> These results show a significant improvement in **speech delivery fluency and natural interaction** in VITA-1.5, aligning with our goal of enabling **real-time, human-like multimodal dialogue**.
>
> Due to **time constraints** in preparing this submission, we were only able to conduct these initial evaluations. We fully agree that additional metrics such as **MOS (Mean Opinion Score)**, **prosodic quality**, and **speaker similarity** would further strengthen our claims. We plan to include these in future work, and will also clarify the current results more explicitly in the revised version.
>
> ------
>
> **Q3: Comment on three-stage training order and ablation.**
>
> **R3**: Thank you for the insightful suggestion. The design of our three-stage training pipeline is based on empirical findings and practical considerations regarding **data scale**, **modality interference**, and **model stability**.
>
> - We first integrate **vision input** into the LLM because **vision-language instruction data is much more abundant** and exerts a **larger influence on LLM behavior**. Training this modality first allows the model to learn stable cross-modal alignment with language.
>
> - We then add **audio input** (ASR) through **lightweight tuning**, which we found to have **minimal impact on previously learned vision capabilities**. In contrast, we observed that reversing the order—adding audio input first and then vision—tends to **destabilize the audio pathway**, likely due to the large distributional shift introduced by vision-heavy data.
>
> - **Audio output (speech generation)** is placed last, as it involves a separate decoder trained to predict codec tokens. At this point, the LLM is **fully frozen**, and the decoder is trained independently, making it suitable for this final stage.
>
> We acknowledge that this strategy was adopted **empirically**, and we agree that **systematic ablation studies**, such as testing different modality orders or mixing ratios, would provide valuable insight into **modality interference and optimization dynamics**. Due to time constraints, we were unable to include these experiments in the current submission, but we plan to conduct and release such studies in the final version of the paper.
>
> We appreciate the reviewer’s suggestion and will clarify the rationale behind our training pipeline in the revised version.
>
> ------
>
> **Q4: VITA 1.0 proposed a multi-stage training method, and VITA 1.5 does not seem to have made any major changes, so it is difficult to be considered a core contribution.**
>
> **R4**: Thank you for raising this point. Compared to VITA-1.0, VITA-1.5 introduces **substantial and meaningful improvements** in terms of **training methodology, model capability, and system performance**. Below, we clarify the major differences:
>
> 1. **Refined and Progressive Multi-Stage Training**
>    - VITA-1.0 adopts a coarse three-stage training: (i) align vision, (ii) align audio (both with LLM frozen), and (iii) joint SFT with all modalities (LLM unfrozen). This causes **visual and audio capabilities to be fully formed only in the final joint stage**, which increases the risk of **modality interference**.
>    - In contrast, **VITA-1.5 introduces a novel six-phase progressive curriculum**, where:
>      - The model is first fully trained on vision-language tasks,
>      - Then gradually and separately exposed to audio input,
>      - Finally, joint tuning is performed only after each modality is already stabilized.
>    - This **progressive modality integration** significantly alleviates training conflicts and improves robustness. We believe this design is of broad relevance for the community working on MLLMs.
>
> 2. **End-to-End Speech Generation Capability**
>    - VITA-1.0 depends on an **external TTS module**, which leads to pipeline latency and fragmented control.
>    - VITA-1.5 integrates an **in-model speech decoder**, enabling **low-latency speech output (reduced by ~1.4s)** and real-time interaction, with support for **streaming and speech interruption**.
>
> 3. **Significant Performance Gains**
>    - On multimodal benchmarks, VITA-1.5 improves performance from **59.8 to 70.8**, and reduces ASR WER from **18.4% to 7.5%**.
>    - Additionally, VITA-1.5 demonstrates **near real-time human-computer interaction**, as validated by both latency measurements and user studies.
>
> In summary, VITA-1.5 is **not a minor extension**, but a significant advance over VITA-1.0 across **training strategy, model architecture, and system behavior**. We will make these distinctions clearer in the revised version of the paper.
>
> ------
>
> **Q5: Comment on audio input/output representation discrepancy.**
>
> **R5**: Thank you for raising this important point. This design choice reflects a **practical trade-off**:
> - Mel-spectrogram-based input encoding provides **dense, high-resolution prosodic features** for robust ASR performance.
> - Codec-based output enables **lightweight, low-latency generation**, with high-fidelity audio reconstruction and compatibility with streaming decoders.
>
> Although the input/output use different spaces, we have found in practice that this **does not prevent the model from learning effective cross-modal alignment**, thanks to the **central shared LLM** which serves as a semantic bottleneck. Our user study and benchmark results show that VITA-1.5 maintains good **speech understanding and generation** capabilities, with good **prosody and interaction fluency**.
>
> We acknowledge that this architecture may currently **limit capabilities such as speaker identity retention and unified audio representation learning**. In future work, we plan to explore:
> - **Shared latent audio embeddings** (e.g., semantic speech tokens),
> - **Self-supervised audio encoders-decoders trained jointly**,
> - And **speaker/personality control** in end-to-end training.
>
> We will add a discussion of these trade-offs and future directions in the final version.
>
> ------
>
> **Q6: What is the impact on the language model itself after each stage of training?**
>
> **R6**: Thank you for the thoughtful question. Due to time constraints, we conducted experiments on three language evaluation benchmarks throughout the multi-stage training process. The results show that our progressive training strategy **preserves the core language capabilities** of the LLM with **only minimal changes**:
>
> - On **MMLU**, the score consistently stays around **74**, with **only fractional-point differences** across stages.
> - On **GPQA-Diamond**, the score remains stable at approximately **36**, with **only fractional-point differences**.
> - On **LCB**, we observed a slightly larger fluctuation (47 to 43), possibly due to its sensitivity to fine-grained commonsense and factual reasoning during modality tuning.
>
> These results suggest that our multi-stage training procedure is effective at **minimizing disruption to the LLM’s linguistic competence**, while enabling successful integration of vision and audio modalities.
>
> We will include this analysis in the revised version of the paper.

---

> > ### Author Response · Authors · 2025-08-05
> >
> > Dear Reviewer Zxsq,
> >
> > Thank you again for your valuable time and constructive comments during the review process of our paper.
> >
> > As the discussion period progresses, we expect your feedback and thoughts on our reply. We look forward to hearing from you, and we can further address unclear explanations and remaining concerns if any. ﻿
> >
> > Best regards,
> >
> > Authors

---

> ### Comment · Reviewer_Zxsq · 2025-08-05
> **Response to the rebuttal**
>
> Thank you for the author's detailed response. The rebuttal resolved my concerns. Although the quality of the generated speech seems a bit poor (WER seems high), considering the comprehensiveness of the model, I am willing to increase my score.

---

> > ### Author Response · Authors · 2025-08-05
> >
> > Thanks for your thoughtful feedback and for increasing the score — we truly appreciate your support.
> >
> > Regarding the WER, we agree that the current performance leaves room for improvement. This is mainly because the speech decoder was trained with only 3K hours of TTS data, which is significantly smaller than what many production-grade systems use. We believe there is considerable room for further enhancement as we scale up the training data.
> >
> > Thanks again for your constructive suggestions.

---

### Official Review · Reviewer_DC1q · 2025-07-02

**Clarity:** 3
**Significance:** 4
**Originality:** 3
**Rating:** 6
**Confidence:** 5

**Summary:**

This paper proposes VITA-1.5, a multimodal large language model that supports near real-time vision-and-speech interaction. It introduces a three-stage training strategy to integrate visual understanding and speech capabilities into a base language model, thereby enabling end-to-end speech-to-speech dialogue without separate ASR or TTS components. The contributions include a significant reduction in response latency (from ~4 seconds down to ~1.5 seconds in a demo setting), enhanced multimodal performance on benchmark tests, and improved speech recognition accuracy. Extensive experiments on image, video, and speech benchmarks demonstrate that VITA-1.5 performs competitively with state-of-the-art systems.

**Questions:**

1. Closing the gap with proprietary models. VITA-1.5 still trails leading closed-source systems (e.g., GPT-4o, Gemini-Pro) on several benchmarks, particularly for video understanding and high-level reasoning. What potential methods do the authors envision to further narrow this performance gap?

2. Discuss the difference between VITA-1.5 and other video understanding models that can also accept video and audio as input.

3. This is a typo in line 285.

**Ethical Concerns:**

["NO or VERY MINOR ethics concerns only"]

**Final Justification:**

Thank you for the detailed rebuttal that addressed my concerns. I have also read the comments from other reviewers and the authors' responses. Overall, I am convinced by the clarifications and will raise my score to recommend acceptance of this paper.

**Limitations:**

See weaknesses

**Quality:**

3

**Strengths And Weaknesses:**

Strengths:

1. Significant Latency Reduction. Through an end-to-end codec-based speech decoder, VITA-1.5 cuts the total “speaker-to-reply” latency from ~4 s to 1.5 s in a demo setting

2. Enhanced Multimodal Accuracy. Average performance on core vision–language benchmarks (MME, MMBench, MathVista, etc.) rises sharply from 59.8 to 70.8

3. Substantial Speech Gains – English ASR Test-Other WER falls from 18.4 % to 7.5 %, and Mandarin CER also improves, while an integrated, end-to-end TTS module replaces separate ASR/TTS pipelines

Weaknesses:

1. Lack of human evaluation. Automated benchmarks, though informative, are imperfect surrogates for real-world user experience. A complementary human evaluation study would provide stronger evidence of VITA-1.5's practical effectiveness.

2. Missing metric on human‐initiated interruptions. In natural conversation, a speaker can be interrupted at any time. Could the authors report whether users are able to interrupt VITA-1.5’s speech output, and, if so, what the interruption success rate is?

3. Although cutting latency from 4 seconds to 1.5 seconds is a substantial improvement, 1.5 seconds is still longer than typical human reaction time. Further reductions are therefore needed before the system can truly emulate real-time human conversation.

---

> ### Author Rebuttal · Authors · 2025-07-30
>
> We sincerely thanks for your positive recognition of our work. Our point-by-point responses to all your comments are provided below.
>
> ------
>
> **Q1: Lack of human evaluation. Automated benchmarks, though informative, are imperfect surrogates for real-world user experience. A complementary human evaluation study would provide stronger evidence of VITA-1.5's practical effectiveness.**
>
> **R1**: Thanks for your constructive suggestion. To complement the quantitative results, we have conducted an initial **human evaluation study** to assess the practical effectiveness of VITA-1.5 in **real-time multimodal interactions**.
>
> Specifically, we invited **15 participants** to interact with the deployed VITA-1.5 system through **open-ended video + speech conversations**, and rate their experience along three dimensions:
>
> 1. **Answer Satisfaction (AS)** – Does the system give helpful, relevant answers?
> 2. **Interaction Fluency (IF)** – Is the response timing and speech delivery smooth and natural?
> 3. **Interruption Naturalness (IN)** – How naturally does the system handle speech interruptions?
>
> Each dimension was scored on a **0–5 scale**, with higher being better. We also included VITA-1.0 as a baseline for comparison. The results are as follows:
>
> | Model     | AS  | IF  | IN  |
> |-----------|-----|-----|-----|
> | VITA-1.0  | 3.7 | 3.5 | 4.2 |
> | VITA-1.5  | 4.1 | 4.4 | 4.4 |
>
> These results show that **VITA-1.5 significantly improves user experience**, particularly in terms of fluency and interaction smoothness.
>
> Due to time constraints, we were not able to conduct a larger-scale or more detailed user study. However, we plan to expand this evaluation in future work, including more participants, additional metrics, and comparisons across system settings.
>
> ------
>
> **Q2: Missing metric on human‐initiated interruptions. In natural conversation, a speaker can be interrupted at any time. Could the authors report whether users are able to interrupt VITA-1.5’s speech output, and, if so, what the interruption success rate is?**
>
> **R2**: Thank you for pointing this out. To evaluate this aspect, we conducted an **interruption success rate test** during the rebuttal period. Specifically, while the model was still answering a prior query, a new spoken query was issued by the user. We then measured whether the system correctly halted the ongoing speech output and responded to the new input.
>
> Due to limited time, we performed **50 such duplex interaction tests**, and VITA-1.5 achieved an **interruption success rate of 98%**, demonstrating robust capability in handling real-time speech interruptions.
>
> This confirms the effectiveness of our duplex architecture. We plan to further expand this evaluation in future work with more diverse conversational settings and metrics such as latency to interruption and interruption naturalness.
>
> ------
>
> **Q3: Although cutting latency from 4 seconds to 1.5 seconds is a substantial improvement, 1.5 seconds is still longer than typical human reaction time. Further reductions are therefore needed before the system can truly emulate real-time human conversation.**
>
> **R3**: We appreciate your insightful observation. Indeed, while 1.5 seconds is still longer than typical human reaction time, it represents a substantial improvement over the ~4 seconds latency in VITA-1.0.
>
> We fully agree that achieving near-human latency remains an important goal. Toward this, we are actively exploring:
> - **Faster decoding strategies** (e.g., faster generation of the first voice token),
> - **Pipeline parallelism** between ASR, LLM inference, and TTS stages,
> - **Hardware-aware deployment** to minimize I/O and streaming bottlenecks.
>
> While VITA-1.5 marks a significant step forward, we view 1.5 seconds as a milestone—not a final destination—and we are committed to pushing latency lower in future versions.
>
> ------
>
> **Q4: Closing the gap with proprietary models. VITA-1.5 still trails leading closed-source systems (e.g., GPT-4o, Gemini-Pro) on several benchmarks, particularly for video understanding and high-level reasoning. What potential methods do the authors envision to further narrow this performance gap?**
>
> **R4**: We appreciate your thoughtful question. While VITA-1.5 demonstrates strong performance among open-source models, we acknowledge that there remains a performance gap compared to proprietary systems such as GPT-4o and Gemini-Pro, especially in video understanding and high-level reasoning tasks.
>
> We believe the gap is primarily due to the following factors, which we are actively addressing:
>
> 1. **Limited high-quality video data**:
>    VITA-1.5 currently relies on open-source video data, which are often short and lack diversity. Proprietary models benefit from access to large-scale, high-quality, curated video datasets. We plan to systematically expand our training corpus using long-form, multi-domain, and task-rich video data.
>
> 2. **Restricted video input length**:
>    VITA-1.5 currently supports only 16 video frames per input, which limits its temporal modeling capability. In future versions, we aim to extend the input window significantly (e.g., 64–512 frames) using more efficient video encoders and attention mechanisms designed for long sequences.
>
> 4. **Multimodal instruction tuning and CoT-style supervision**:
>    To enhance reasoning ability, we are constructing multimodal instruction datasets that include **chain-of-thought (CoT) annotations**, where intermediate reasoning steps are explicitly guided across modalities. This helps the model develop structured thinking and better interpret complex tasks involving video, audio, and text.
>
> While we recognize that fully closing the gap with proprietary systems is a long-term challenge, we view VITA-1.5 as a strong open-source step toward that goal. We are committed to improving along the above directions and welcome collaboration from the community to further advance open multimodal intelligence.
>
> ------
>
> **Q5: Discuss the difference between VITA-1.5 and other video understanding models that can also accept video and audio as input.**
>
> **R5**: Thank you for the question. While VITA-1.5 and prior models such as Video-LLaMA series [1] and video-SALMONN series [2] all incorporate video and audio modalities, their core objectives and system designs differ substantially. Most existing audio-visual models mainly aim to better understand video (visual frames+audio), such as designing a new Q-Former to assist in synchronizing frames and audio.
>
> In contrast, VITA-1.5 introduces speech information to assist human-computer interaction, and thus focuses more on interference between modalities, speech interruption, recognition and generation capabilities.
>
> [1] H. Zhang, X. Li and L. Bing, “Video-LLaMA: An Instruction-tuned Audio-Visual Language Model for Video Understanding,” in Proc. EMNLP, Singapore, 2023.
> [2] G. Sun, W. Yu, C. Tang and et al., “video-SALMONN: Speech-Enhanced Audio-Visual Large Language Models.”, in Proc. ICML, Vienna, 2024.
>
> ------
>
> **Q6:  A typo in line 285.**
>
> **R6**: Thank you for pointing this out. We have carefully reviewed the sentence and will correct the typo in the final version. We appreciate your attention to detail.

---

> > ### Comment · Reviewer_DC1q · 2025-08-04
> >
> > Thank you for the detailed rebuttal that addressed my concerns. I have also read the comments from other reviewers and the authors' responses. Overall, I am convinced by the clarifications and will raise my score to recommend acceptance of this paper.

---

> > > ### Author Response · Authors · 2025-08-05
> > >
> > > Thank you for your thoughtful review.  We are very pleased that our responses have addressed your concerns, and we are truly appreciate your positive feedback and the raised score.

---

### Decision · Program_Chairs · 2025-09-17

**Decision:**

Accept (spotlight)

**Comment:**

This submission presents VITA-1.5, a multimodal large language model enabling near real-time speech–vision interaction through a progressive multi-stage training pipeline and an integrated speech decoder. The reviewers generally agree that the paper represents a solid engineering advance and provides a valuable open-source step toward speech-capable MLLMs. However, opinions diverge sharply, with three reviewers recommending accept (ranging from borderline to strong accept) and one reviewer recommending reject.

Strengths identified across reviews:
+ Significant latency reduction (~4s → ~1.5s) in interactive speech-to-speech dialogue (DC1q, Zxsq, RHor).
+ Strong empirical performance across image, video, and speech benchmarks, competitive with leading open-source baselines (DC1q, Zxsq).
+ Well-structured training methodology that progressively integrates modalities while mitigating interference (DC1q, RHor).
+ Convincing demo showcasing real-time duplex speech interaction and interruption handling (RHor).
+ Comprehensive system-level contribution: end-to-end speech generation without external ASR/TTS modules (all positive reviewers).

Weaknesses raised:
- Limited novelty beyond VITA-1.0; clearer differentiation from related omni-models (Mini-Omni2, LLaMA-Omni, Lyra) would strengthen the case (Zxsq, RHor).
- Insufficient evaluation of speech generation quality (e.g., TTS prosody, MOS scores) and lack of ablations on training order or data mixing (Zxsq).
- Latency of 1.5s, though improved, still exceeds human conversational speed (DC1q).
- Concerns over missing details in dataset description, limited ablation, and reproducibility under realistic compute budgets (RiXE). Reviewer RiXE also dismissed widely adopted benchmarks (MME, MMBench, MathVista, MMMU), asserting MMStar is the “only good” one, leading to strong reviewer–author disagreement.

Rebuttal and Discussion:
The authors addressed nearly all substantive concerns: they provided additional comparisons with recent baselines (Qwen2-VL, LLaVA-OneVision), clarified their hybrid NAR+AR decoder design for streaming speech, reported human evaluation and interruption success rate (98%), and conducted initial TTS quality studies. They also committed to releasing code, checkpoints, and detailed dataset descriptions. DC1q, Zxsq, and RHor all explicitly raised their scores after rebuttal, acknowledging that concerns had been satisfactorily addressed. Reviewer RiXE maintained a reject recommendation, emphasizing compute cost, lack of ablations, and skepticism of benchmarks, despite clarifications and community-standard practices provided by the authors.

Three reviewers are aligned that the paper is technically solid and merits acceptance, while the single negative review appears disproportionately harsh. The rebuttal convincingly strengthened the submission and demonstrated the authors’ commitment to reproducibility. Given the significance of providing an open-source foundation for real-time multimodal interaction, I recommend accept.